# Relative Telomere Length Is Associated with the Risk of Development and Severity of the Course of Age-Related Macular Degeneration in the Russian Population

**DOI:** 10.3390/ijms241411360

**Published:** 2023-07-12

**Authors:** Olga P. Dmitrenko, Olga I. Abramova, Nataliia S. Karpova, Malik K. Nurbekov, Ekaterina S. Arshinova

**Affiliations:** 1Laboratory of Regulation of Reparative Processes, Federal State Budgetary Institution “Research Institute of Pathology and Pathophysiology”, 125315 Moscow, Russia; dolga6528@gmail.com (O.P.D.); nataliiakarpova.sp@gmail.com (N.S.K.);; 2Department of Ophthalmology, Federal State Budgetary Educational Institution of Continuing Professional Education “Russian Medical Academy of Continuing Professional Education” of the Ministry of Health of the Russian Federation, 125993 Moscow, Russia

**Keywords:** aging, age-associated diseases, age-related macular degeneration, relative telomere length, early diagnosis biomarker, biomarker for predicting disease progression

## Abstract

One of the most significant factors for age-related macular degeneration (AMD) development is considered to be aging, the processes of which are closely associated with telomere shortening. The different forms, indicators of aggressiveness, and intensities of AMD can be observed in the same age group, confirming the need to find a biomarker for early diagnosis and be capable of monitoring the progression of the pathological process. Therefore, we investigated whether the relative telomere length (RTL) has any connection with the risk of development of disease and its progression. RTL was measured using RT-PCR in 166 people, including 96 patients with AMD. RTL was significantly lower in patients with AMD. Women were more likely to develop AMD than men (odds ratio (OR) = 9.53 × 10^6^ vs. OR = 1.04 × 10^8^, respectively). The decrease in RTL in patients reliably correlated with the progression of AMD, and the smallest RTL was observed in late-stage patients. RTL < 0.8 is a significant risk factor for disease progression. The results of our research showed that RTL may be considered as a potential biomarker and a promising predictor of disease progression in patients with early AMD.

## 1. Introduction

Age-related macular degeneration (AMD) is a chronic progressive multifactorial disease that can be caused by genetic, environmental, and constitutional factors. AMD affects the macular area of the retina, which leads to irreversible vision loss in the elderly [1,2]. According to WHO data, 8.7% of the global population is affected by AMD, and by 2040 the number of patients is expected to increase to 288 million [3]. In Russia, according to various data, up to 46% of the population over the age of 65 suffer from AMD [4]. The EpiCast report on age-related macular degeneration indicates that the majority of the newly diagnosed AMD cases in 2016 occurred in women (65.76%) [5]. The Blue Mountains study previously demonstrated that the incidence of specific early AMD lesions and late AMD was greater in women compared to men [6]. Despite the fact that AMD is a disease of the elderly, a recent study by Brandl et al. (2022) found early signs of AMD in subjects aged 25. The results of this study showed that 22.7% of patients aged 35–55 with an early stage of AMD at the initial stage progressed to a late stage of AMD within 18 years of observation [7].

The pathological development and progress of AMD are contributed to by multiple environmental and lifestyle factors such as gender, ethnicity, smoking, being overweight, diabetes mellitus, atherosclerotic arterial disease, diet, etc. [8]. In their recent research, Hunt et al. (2022) confirmed previous reports that risk factors such as sex and ethnicity are strongly correlated with AMD [9]. Moreover, other studies demonstrated that lifestyle changes such as smoking cessation, exercising, and following a Mediterranean diet with food supplements lower the risk of developing AMD [10,11]. Moreover, age is one of the most important risk factors for the development of AMD, since the senescence-related processes linked to it alter the molecular pathways involved in the pathological development. As telomeres play an important role in the protection and support of genomic and cellular wholeness, the telomeres’ length and shortening through the years is seen as a one of the hallmarks of aging [12,13]. Telomeres are repeated nucleotide structures (TTAGGG) located at the ends of the eukaryotic linear chromosomes, which protect the terminal regions of chromosomal DNA from degradation [14]. An incomplete chromosome replication during cell division and destructive factors such as nucleases, oxidative processes, and free oxygen radicals can damage telomeres. This damage results in telomeres becoming shorter [15]. Environmental stress factors, an unhealthy lifestyle, and exposure to UV light can also influence the telomere length change [16,17].

The replication of linear DNA leads to the so-called end replication problem, when the DNA replication mechanism is not able to completely reproduce the ends of the chromosomes itself. As a result, 50–200 bp is lost in each S phase in cultured human cells’ telomeric DNA [18]. The gradual shortening of the telomeres in the replicating cells causes the depletion of the proliferative potential of these cells and may be associated with oxidative stress, mitochondrial dysfunction, and inflammation, which play a fundamental role in the pathogenesis of age-associated diseases, including AMD [19,20,21,22,23]. Despite the fact that the length of telomeres varies in different types of tissues, the data show a strong correlation of telomere length with age in different somatic tissues [19,20].

Recent studies demonstrated the inverse correlation between RTL and the development of heart disease, type two diabetes, Alzheimer’s disease, Parkinson’s disease, and various forms of oncological disorders [14,24,25]. However, the contribution of telomere length to the development of AMD has not been sufficiently estimated. The available literature on the relation of telomere length to the AMD is limited and contradictory [22,23,26,27]. The study of telomere dysfunction in AMD would make it possible to take a fresh look at early diagnosis and the possibility of predicting the progression of the pathological process. This study aims to evaluate the relationship between telomere length and AMD and to analyze if the relative length of telomeres depends on the stage of the disease.

## 2. Results

The characteristics of the groups are shown in Table 1. There were no significant differences in gender and age between patients with AMD and the control group (*p* > 0.05). There were significant differences between the study groups in terms of body mass index > 30, cardiovascular disease, diabetes mellitus, physical inactivity, and RTL. Statistically significant differences in the distribution of long and short telomeres between the studied groups were identified. There were more patients with short telomeres in the AMD group than in the control group (*p* < 0.00001). 

Initially, we calculated models of logistic regression without including telomere length. None of the models, without including telomere length, were statistically significant. However, models that included such individual covariates as sex, physical inactivity, and cardiovascular disease showed statistical significance, with the complete regression model remaining statistically insignificant. The highest sensitivity (87%) and specificity (58%) obtained are shown in Appendix A (Appendix A).

Then, we included the relative length of telomeres in each model. The inclusion of telomeres in the models made all the results statistically significant. However, when calculating model 4, which included such covariates as age, sex, current smoking, diabetes, cardiovascular disease, hypertension, obesity, and physical inactivity, within the stratified sex samples the software stats (version 3.6.2) packages for R 4.0.4 (R Foundation for Statistical Computing, Vienna, Austria) produced a calculation error. This error was due to the fact that in the control group there were only two people with diabetes; their number was not enough to correctly calculate the model, so we had to exclude it.

The probability of AMD development increased in the model by age and sex (OR = 6.96; 95% CI 4.15 × 10^4^–3.26 × 10^9^; *p* < 0.0001). This correlation remained stable with the addition of other AMD risk factors (current smoking, cardiovascular disease, and physical inactivity), maintaining an independent link between RTL and AMD. We found that the shorter telomere length was associated with increased chances of AMD development in women (OR = 9.53 × 10^6^; 95% CI 2.38 × 10^4^–1.64 × 10^10^; *p* < 0.0001) rather than in men (OR = 1.04 × 10^8^; 95% CI 2.02 × 10^3^–2.33 × 10^15^; *p* = 0.007) in the age-adjusted logistic regression model. This correlation remained significant in all other models of AMD (Table 2, Figure 1). The model sensitivity and specificity are presented in Appendix A.

The analysis of the severity of the course of AMD determined that the smallest relative length of telomeres was observed in patients with late AMD compared with patients with early AMD and patients in whom AMD was absent (*p* < 0.00001) (Table 3 and Figure 2). 

The correlation analysis showed a statistically significant correlation between RTL and the severity of the disease, both when comparing RTL across all groups and in pairs (Table 4). The decrease in RTL in patients reliably correlated with the progression of AMD (r = −0.726; *p* < 0.00001).

In AREDS subgroups 2 and 3, the relative telomere length was ≥0.8 in 15 patients and <0.8 in 50 patients. The absolute and relative risks and the change in the risk of AMD severity were assessed, considering the smallest relative length of telomeres observed in late AMD patients. The results of the statistical univariate prediction of the AMD severity using Pearson’s chi-square test showed that relative telomere length <0.8 is a statistically significant risk factor for disease severity (*p* < 0.00001) (Table 5 and Figure 3).

## 3. Discussion

We analyzed the correlation between the risk of developing AMD and the relative length of telomeres to determine the dependence between telomeres and AMD. Our results showed that patients with AMD had relative telomere lengths that were significantly lower than those in the control group. The probability of AMD development in individuals with short telomeres increased in the age- and sex-adjusted model, and the relationship remained significant in AMD risk-factor-adjusted models. Women had a better chance of developing AMD than men (OR = 9.53 × 10^6^ vs. OR = 1.04 × 10^8^). We also found that the decrease in RTL in patients reliably correlated with AMD progression and that the smallest relative lengths of telomeres were observed in patients with late AMD. In assessing the change in the risk of severity, relative telomere length <0.8 was a significant risk factor for disease severity. Thus, RTL may be considered as a potential biomarker and a promising predictor of disease progression in patients with early AMD.

AMD is the primary cause of vision loss. Various forms, different indicators of aggressiveness. and the intensity of AMD can be observed within the same age group. Late onset is known to be associated with the worst prognosis. Large-scale population studies showed that 19–28% of unilateral cases of any AMD become bilateral. Patients with advanced AMD risk experiencing damage to the second eye within 5 years in 27–68% of cases [28].

Previous studies found that shorter telomere lengths are associated with increased chances of AMD development, which is consistent with our results [22,23]. Weng et al. (2015) found that older people have a significant relationship between AMD and RTL (OR = 2.24; 95% CI = 1.68–3.07; *p* = 0.0001) after adjusting for age and sex. The results of this study showed a very significant correlation between the late stage of AMD (geographical atrophy) and RTL (OR = 4.81; 95% CI = 3.15–7.82; *p* = 0.0001) after adjusting for age and sex [22]. Koller et al. (2022) found a link between shorter relative telomere lengths and AMD in older women, regardless of age and other potential risk factors (OR = 0.915 in women vs. OR = 0.892 in men; *p* = 0.003) [23]. A comparison of RTL between persons without AMD and those with early or late AMD show a very significant difference in telomere length (*p* < 0.001). The shortest RTL was found in late AMD (0.86 ± 0.17) compared to early RTL (0.88 ± 0.17). Banevicius et al. (2021) found longer telomeres in patients with early-stage AMD when compared to the control group in the Lithuanian population (T/S, median (IQR): 1.207 (1.319) vs. 0.778 (1.057); *p* < 0.001) [27]. However, there was no difference in telomere length between patients with early and late AMD (T/S, median (IQR): 1.207 (1.319) vs. 0.918 (1.406); *p* = 0.064). According to a study by Immonen et al. (2013), no differences were observed in comparing the average (SD) telomere length in patients with progressive AMD compared to the healthy group (SD: 0.68 vs. 0.69; *p* = 0.485) [26]. There was also no correlation between telomere length and risk factors (age, sex, smoking status, type two diabetes, and treatment of hypertension).

Conflicting results may be due to differences in the studied populations, research design, measurement methods, and statistical modeling [29]. Since only older people (over 70 years of age) were involved in these studies, it is difficult to say whether shorter telomeres increase the risk of AMD development or whether they only serve as a marker of biological aging. Many eye diseases, including AMD, affect both of a patient’s eyes. The inclusion of one or both eyes is thought to affect the statistical analysis of the study data, as the outcome indicators for both eyes of a participant are usually positively correlated [30,31]. It is not known what design was used in earlier studies [22,23,26,27]. It is possible that the researchers chose the design entirely based on the patent or the most severely affected eye in the patient.

The main limitations of this study are the small sample size of the studied groups and the measurement of telomere length at only one point in time. For future studies, it is important to consider that a larger sample size will provide more comparable data for the population and increase statistical relevance. A longitudinal study with consecutive RTL measurements at different points in time would assess the impact of the telomere-length depletion rate on the disease progression rate. The strengths of the study are that all participants were recruited and examined by an experienced ophthalmologist using the same standardized procedures. The study also presents a wide age range, covering persons aged 46–88 years, and, when assessing the severity of AMD, we used a two-eye design for different groups, in which each eye was evaluated as a separate case. 

It should be noted that the DNA in our study was extracted from buccal epithelium cells. This method was preferable to venous blood because it is less invasive and does not affect the evaluation of RTL [19,32]. In addition, studies showed that DNA from buccal scraping samples has comparable quality with DNA from blood samples, and telomere length analysis from buccal epithelial cells is statistically significant [33,34,35]. 

## 4. Materials and Methods

### 4.1. Study Population

This study was carried out on a sample of persons of the Caucasian population over 45 years of age, who passed the standard ophthalmological examination in the ophthalmological department of the State Medical University named after S.P. Botkin DZM G. in Moscow in 2019–2021. The Ethics Committee of the Federal State Budgetary educational institution of continuing professional education, the “Russian Medical Academy of Continuing Professional Education” of the Ministry of Health of the Russian Federation, approved the research protocol. Each participant gave written and informed consent in accordance with the Helsinki Declaration. 

AMD diagnosis was established according to the recommendations of the American Academy of Ophthalmology (AAO) and based on the criteria of the Russian national clinical guidelines for “Age-related macular degeneration” [1,2]. Patients with acute and chronic diseases in the stage of exacerbation of the visual organs, glaucoma, uveitis of various etiologies, complete complications by cataract, retinal detachment, iris rubeosis, and autoimmune and oncological processes of any localization were excluded from the study (*n* = 2). Also excluded from the study were participants who did not agree to a beech scraper (*n* = 6). In the analysis phase, 23 participants were excluded because of insufficient DNA (*n* = 14) or standard deviation (SD) of Ct ≥ 1 for telomeres or single copy gene (SCG) (*n* = 9). The final sample size was 166, including 96 patients with AMD.

To analyze the severity of AMD, we analyzed the data of each eye as a separate case. Subsequently, the patients’ eyes were divided into 4 groups depending on changes in the retina according to the Age-Related Eye Disease Study (AREDS) classification [30,36]. Based on the characteristics of each eye after the examination, group 1 (AREDS 1) included 143 eyes in which no changes in characteristic AMD were detected. Group 2 (AREDS 2) included 54 eyes with many small drusen, a few medium-sized drusen (63–124 µm), and/or pigmentary changes in one or both eyes. Group 3 (AREDS 3) had 44 eyes with many medium-sized drusen or one or more large drusen (>125 µm) in one or both eyes. Group 4 (AREDS 4) is characterized as the most severe course of the disease and is expressed as an advanced dry form (i.e., geographic atrophy involving the macula) or an exudative form with choroidal neovascularization in one eye. This group included 91 eyes. 

### 4.2. Genomic DNA Extraction and Real-Time Polymerase Chain Reaction (RT-PCR) for Relative Telomere Length (RTL) 

All samples were collected from all AMD patients prior to treatment. DNA was extracted from buccal epithelial cells and purified with the genomic DNA extraction kit (Evrogen LLC, Moscow, Russia), in accordance with the instructions of the manufacturer. The high-molecular DNA was stored at −20 °C. The quantity and quality of the isolated DNA was assessed in the NanoDrop 1000 spectrophotometer in accordance with accepted standards. Telomere length analysis was performed by real-time PCR on a CFX 96 programmable amplifier (Bio-Rad, Hercules, CA, USA) according to the original protocol taken from the literature (Cawthon, 2009), using specific primers synthesized at Evrogen LLC, Russia (Table 6) [37]. 

The reaction mixture for telomere analysis contained 20 ng genomic DNA, 1X qPCRmix-HS (Evrogen LLC, Russia) telomere primer pair telg and telc (final concentrations 900 nM each), beta-globin primer pair hbgu and hbgd, (final concentrations 500 nM each), and 0.75 × SYBR Green I in final volume 25 µL. The thermal cycling profile was Stage 1: 15 min at 95 °C; Stage 2: 2 cycles of 15 s at 94 °C and 15 s at 49 °C; and Stage 3: 32 cycles of 15 s at 94 °C, 10 s at 62 °C, 15 s at 74 °C with signal acquisition, 10 s at 84 °C, and 15 s at 88 °C with signal acquisition. The 74 °C reads provided the Ct values for the amplification of the telomere template; the 88 °C reads provided the Ct values for the amplification of the beta-globin. For each sample and standard, there were three repeats of each telomeric and SCG reaction. 

All obtained data were analyzed using CFX Manager TM software version 3.1 (Bio-Rad). The relative length of telomeres was estimated by the T/S index, which was calculated as the ratio of the number of copies of telomeric repeats to the number of copies of the reference gene.

### 4.3. Statistical Analysis

Statistical analysis was performed using SPSS 17.0 (SPSS, Chicago, IL, USA) and R 4.0.4 (R Foundation for Statistical Computing, Vienna, Austria). The normality of the data distribution was checked using the Shapiro–Wilk test. Data are presented as mean ± standard deviation (SD) and using absolute numbers with percentages. The data on the relative length of telomeres are presented as mean ± SD. Differences in age between groups were analyzed using Student’s t-test. The Mann–Whitney test was used to compare the relative telomere length between the two groups. The gender and distribution of AMD stages are reported using absolute numbers with percentages. The distribution of long and short telomeres in AMD and control groups was compared using the test χ^2^. The Kruskal–Wallis criterion was used to check the differences in the mean values of the relative telomere length in the subgroups. Correlation was assessed using Spearman’s and Kendall correlation coefficients. Odds ratios (ORs) were calculated using logistic regression models, taking into account possible factors influencing both telomere length and disease development, to estimate the relationship between telomere length and BMD. A one-factor forecasting model was used to assess absolute risks, risk change, and relative risk of AMD severity. The level of significance was considered significant at *p* < 0.05.

## 5. Conclusions

In this study, we found that the shortness of the relative telomere length in patients significantly correlated with the progression of AMD. The most reduced RTL was observed in patients at an advanced stage of pathology. It was determined that RTL < 0.8 is a significant risk factor for disease progression. The results of our study show that RTL may be considered as a potential biomarker and a promising predictor of disease progression in patients with early AMD. However, a longitudinal study on a larger sample size with sequential measurements of RTL at different time points is required to evaluate the effect of the rate of telomere length depletion on the rate of disease progression.

## Figures and Tables

**Figure 1 ijms-24-11360-f001:**
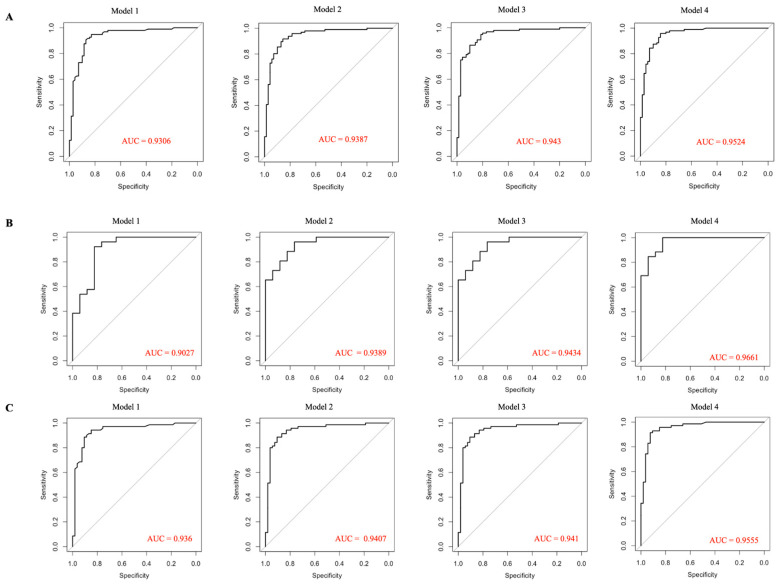
ROC curves for logistic regression models with RTL data: for all (**A**), only for women (**B**), and only for men (**C**). Model 1 receiver operating characteristic (ROC) curve adjusted for RTL. Model 2 ROC curve adjusted for age and sex. Model 3 ROC curve adjusted for age, RTL, and current smoking. Model 4 ROC curve adjusted for RTL, cardiovascular disease, and physical inactivity. Area under the curve (AUC) values are the probability that a randomly selected “case” (here, an AMD) is ranked as being at greater risk of being a “case” than a randomly selected control of the same covariates.

**Figure 2 ijms-24-11360-f002:**
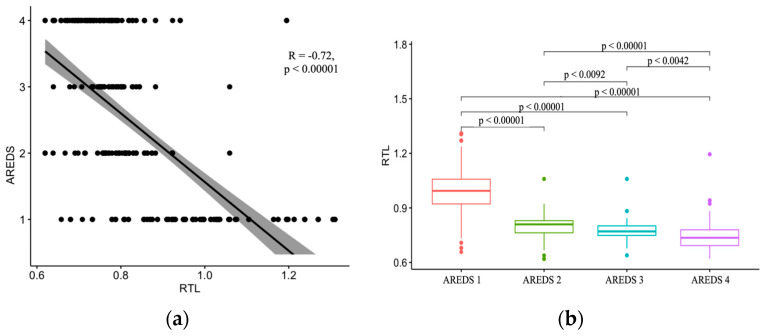
Scatter plot (**a**) and boxplot (**b**) for RTL grouped by AREDS 1–4 with pairwise comparisons; *p*-value according to Kruskal–Wallis statistical test. The points in Figure 2a correspond to the number *n*, while the individual points in Figure 2b are outliers that are shown by default. If the value is outside the 1.5 interquartile range from the nearest quartile, then it is considered abnormal.

**Figure 3 ijms-24-11360-f003:**
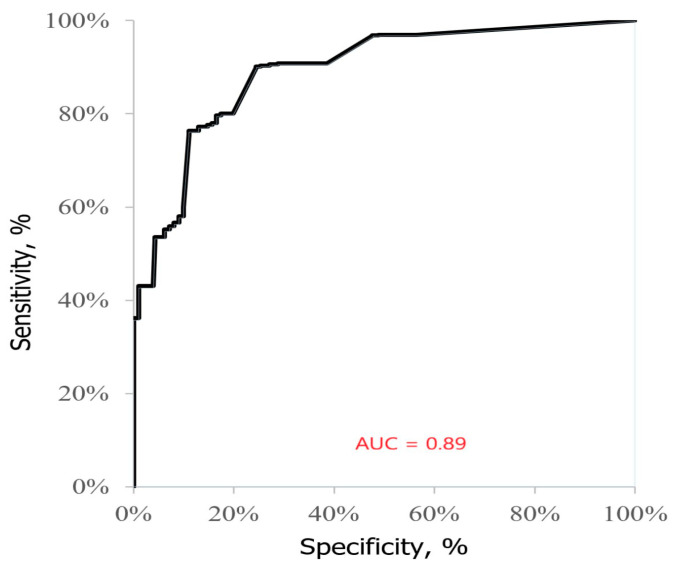
ROC curves for AMD severity prediction model.

**Table 1 ijms-24-11360-t001:** Characteristics of the study groups.

Characteristics	Group		*p* Value
	AMD, *n* = 96	Control, *n* = 70	
Age, years, mean ± SD	70.96 ± 9.72	69.23 ± 11.91	0.459
45–59 years, *n* (%)	18 (18.75)	13 (18.57)	0.969
60–74 years, *n* (%)	38 (39.58)	29 (41.43)	
75–90 years, *n* (%)	40 (41.67)	28 (40.0)	
Sex, *n* (%)			
female	71 (73.96%)	48 (68.57%)	0.447
male	25 (26.04%)	22 (31.43%)	
Smoking, *n* (%)	13 (13.54%)	11 (15.71%)	0.694
Body mass index > 30, *n* (%)	29 (30.21%)	9 (12.86%)	0.009
Hypertension, *n* (%)	87 (90.62%)	62 (88.57%)	0.667
Cardiovascular disease, *n* (%)	66 (68.75%)	32 (45.71%)	0.011
Diabetes mellitus, *n* (%)	12 (12.50%)	2 (2.86%)	0.027
Physical inactivity, *n* (%)	46 (47.92%)	8 (11.43%)	<0.0001
Relative telomere length, mean ± SD	0.77 ± 0.08	0.99 ± 0.13	<0.0001
Relative telomere length, median (IQR)	0.76 (0.09)	0.99 (0.11)	<0.0001
Long telomeres, *n* (%) *	2 (2.1)	34 (48.6)	<0.00001
Short telomeres, *n* (%) *	94 (96.9)	36 (51.4)	

Descriptions for Table 1: * RTL below the median telomere length of the control group (median = 0.9938) was considered as “short telomeres”, and RTL above the median telomere length of the control group was considered as “long telomeres”.

**Table 2 ijms-24-11360-t002:** Logistic regression analysis of RTL and AMD.

RTL	All, *n* = 166		Women, *n* = 119		Men, *n* = 47	
	OR (95% CI)	*p* Value	OR (95% CI)	*p* Value	OR (95% CI)	*p* Value
Model unadjusted	1.37–1.87	0.045	1.32 (0.93–1.89)	0.127	1.53 (0.84–2.87)	0.173
Model 1 adjusted for sex	8.46 × 10^7^ (7.94 × 10^5^–2.55 × 10^10^)	<0.0001	—		—	
Model 2 adjusted for age and sex	6.96 × 10^6^ (4.16 × 10^4^–3.26 × 10^9^)	<0.0001	9.53 × 10^6^ (2.38 × 10^4^–1.64 × 10^10^)	<0.0001	1.04 × 10^8^ (2.02 × 10^3^–2.33 × 10^15^)	0.007
Model 3 adjusted for age, sex, and current smoking	2.74 × 10^7^ (9.93 × 10^4^–2.56 × 10^10^)	<0.0001	9.48 × 10^6^ (2.36 × 10^4^–1.63 × 10^10^)	<0.0001	1.66 × 10^9^ (8.09 × 10^3^–4.28 × 10^17^)	0.006
Model 4 adjusted for sex, cardiovascular disease, and physical inactivity	6.00 × 10^7^ (4.17 × 10^5^–2.96 × 10^10^)	<0.0001	3.00 × 10^7^ (1.46 × 10^5^–2.59 × 10^10^)	<0.0001	1.66 × 10^9^ (8.09 × 10^3^–4.28 × 10^17^)	0.014

Descriptions for Table 2: OR—odds ratio; 95% CI—95% confidence interval.

**Table 3 ijms-24-11360-t003:** Characteristics of patients depending on the severity of AMD.

Characteristic	Stage				*p* Value
	Absence of AMD (AREDS 1), *n* = 143	Early Stage (AREDS 2), *n* = 54	Intermediate Stage (AREDS 3), *n* = 44	Late Stage (AREDS 4), *n* = 91	
Men, *n* (%)	35 (24.47)	20 (37.03)	7 (15.9)	24 (26.37)	0.115
Women, *n* (%)	108 (75.53)	34 (62.97)	37 (84.1)	67 (73.63)	
Age, years, mean ± SD	68.85 ± 12.04	70.22 ± 9.97	71.64 ± 9.09	71.71 ± 9.29	0.318
Relative telomere length, mean ± SD	0.99 ± 0.13	0.80 ± 0.08	0.77 ± 0.06	0.75 ± 0.08	<0.00001

Descriptions for Table 3: *n*—number of eyes.

**Table 4 ijms-24-11360-t004:** Correlation analysis between RTL and AMD severity by AREDS group.

	Spearman’s Correlation Coefficient (rho)	*p* Value	Kendall Correlation Coefficient (tay)	*p* Value
All AREDS groups	−0.726	<0.00001	−0.596	<0.00001
AREDS 1-AREDS 2	−0.603	<0.00001	−0.496	<0.00001
AREDS 1-AREDS 3	−0.615	<0.00001	−0.505	<0.00001
AREDS 1-AREDS 4	−0.720	<0.00001	−0.582	<0.00001
AREDS 2-AREDS 3	−0.266	0.008	−0.219	0.009
AREDS 2-AREDS 4	−0.367	<0.00001	−0.302	<0.00001
AREDS 3-AREDS 4	−0.249	0.004	−0.204	0.004

**Table 5 ijms-24-11360-t005:** Assessment of absolute risk, change in risk, and relative risk of severity of AMD.

Factor	Factor Frequency, *n* (%)		Risk Change (95% CI)	Relative Risk (95% CI)	*p* Value
	No Factor	There Is a Factor			
Relative telomere length < 0.8	15 (14.7%)	50 (78.1%)	63.4 (51.2–75.7)	5.31 (3.27–8.63)	<0.0001

**Table 6 ijms-24-11360-t006:** Primer sequences for RT-PCR.

Primer Name	Sequences
telg	ACACTAAGGTTTGGGTTTGGGTTTGGGTTTGGGTTAGTGT
telc	TGTTAGGTATCCCTATCCCTATCCCTATCCCTATCCCTAACA
Hbgu	CGGCGGCGGGCGGCGCGGGCTGGGCGGcttcatccacgttcaccttg
Hbgd	GCCCGGCCCGCCGCGCCCGTCCCGCCGgaggagaagtctgccgtt

## Data Availability

Not applicable.

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
