# Peer review of "Relative Telomere Length Is Associated with the Risk of Development and Severity of the Course of Age-Related Macular Degeneration in the Russian Population"

_ijms, 2023, doi:10.3390/ijms241411360_

Round 1

Reviewer 1 Report

The authors should organize the article more carefully to avoid too many careless mistakes. The comments are included in the word file, please check.

Overall the presentation is readable. just there are some vocabularies need to be corrected, for example persons.

Author Response

We are grateful for your valuable and extremely useful comments, which allow us to significantly improve the article.

Below are answers to questions that arose during the review:

  1. In terms of writing, there are two points could be corrected to improve the article.
    a.) First, every abbreviation needs to be described when it is used in the first time and consistently used the abbreviation in the following paragraph. For example, in line 12, AMD was not described. So did “OR” in line 19.
    b.)
    Second, the decimal point. The decimal point should be “.”. in the article, the authors intend to use comma instead of decimal point, which need to be corrected.  

Answers:

  1. We added a description of each abbreviation when we first used it.
  2. In the Russian Federation, it is customary to use a comma to separate the integer and fractional (decimal) parts of a number. Taking into account your remark, we corrected the spelling of numbers and replaced the comma with a dot.

  1. In the result tables and figures, the legends have mistaken in multiple sites, please check thoroughly.
    a.) The description under the table 1 is the “Descriptions for Table 2”. Moreover, the statement “P value is greater ≤ 0.05” is confusing.
    b.) The legend of Figure 1: There is no Model 4 in the figure.
    c.) Table 4: there are no star mark in the Table.

Answers:

  1. We corrected that mistake.
  2. We corrected that mistake.
  3. Since the symbol "*" was hard to see, we marked this item as "All AREDS groups" and removed descriptions in table 4.

  1. In the introduction part, the author need to provide more information such as, the known risk factors of AMD (assume are the factors in table 1, but not described in the text yet), the correlation of AMD and previous reported markers.

Answer:

Information on risk factors for the development of AMD is included in the introduction part.

  1. The study established 5 model to analyze the correlation between AMD and the related factors. However, the model 4 was loss and only described “the program made a mistake” what is the mistake leads to the exclusion of this model need to be specifically described. 

Answer:

The mistake was due to the fact that in the control group there were only 2 people with diabetes and their number was not enough to correctly calculate the model. We have added this explanation to the article.

  1. The models which adjusted different factors in inconsistent between the Table 2 and Figure 1. Please check if it is typo, or the analysis is different. Is the latter is the situation, the result writing should be separated. 

Answer:

We have corrected the designation of the models in figure 1 to the correct ones.

  1. In the results, some parameters does not described clearly, for example, what is the sensitivity and the specificity? what is the compared group of the p value in Table 3? The detailed of these parameters are required. 

Answer:

Model sensitivity and specificity are presented in Supplement 2.

In Table 3, a comparison of patient characteristics depending on the severity of AMD was performed using the Kruskall-Wallace test. This information is in the materials and methods section.

  1. In table 3, the sample is based on number of eyes. However, the telomere length is analysis from the buccal epithelial cells. Therefore, the n number must not be the same. How do you deal with the patient with AMD in 2 eyes? Also, the n number seems controversial with the “two-eye design” in line 198. 

Answer:

Lee et al. (DOI: 10.1016/j.ophtha.2011.09.025) divided ophthalmic study designs into 4 types: 1-eye design, 2-eye design, double-eye design, and subject design. The dual-eye design includes a distribution of scores for both eyes of each participant within the study and is categorized as a single-group design, a multi-group design, and a mixed design. In a multi-group design, the participant's 2 eyes are assigned to different groups.

Based on the characteristics of each eye after the ophthalmological examination, we assigned each eye separately to 4 different AREDS groups. For example, one patient did not have AMD in her left eye, but her right eye was AREDS 4.

Buccal scraping of the epithelium was used because it is a minimally invasive source of biological material. This method was preferred over venous blood as it is less invasive and has a similar result in RTL analysis [PMCID: PMC3615479, PMCID: PMC5723637]. The use of diseased tissue (RPE) to measure telomere length is not possible, as it requires serious intervention, and patients will not agree to this procedure. Since both eyes belong to the same patient, we can use the same telomere value for each eye to assess the severity of AMD.

In the event that we were to conduct a longitudinal study with an ophthalmic examination and sequential RTL measurements at different time points, a design with 2 eyes for different groups would not be acceptable. For example, a patient with no AMD in one eye at baseline showed signs of early AMD after 3 years. In this case, a dynamic change in the length of the telomeres of the buccal scraping cells would be noted. In this case, we would use the design of the most severely affected eye in the subject.

We corrected line 198.

  1. In Figure 2a, what is the AREDS 0, in the text, only AREDS 1-4 are mentioned, please check and reorganized. In Figure 2b, what is the dots in the figure mean? The n numbers you mentioned in Table 3 are way more than the dots on the Figure 2b, please describe clearly for the meaning of these dots. 

Answer:

The points in figure 2a correspond to the number n, while Individual points in 2b are outliers that are shown by default. If the value is outside the 1.5 interquartile range from the nearest quartile, then it is considered abnormal.

  1. in the telomere biology, RTL have less correlation with other telomere length measurements such as the gold-standard Telomere Restriction Fragment (TRF) assay, quantitative-FISH, flow-fish and ddPCR based measurements. Do you have any data prove that the RTL assay is correlate to other telomere length measurements? 

Answer:

Telomere length analysis  according to the original protocol taken from the literature (Cawthon, 2009). Cawthon points out in the original article that the results of the method he developed coincide with the results of TRF analysis. Correlation of relative T/S ratios measured by MMQPCR and mean TRF lengths determined by Southern blot analysis, in whole blood DNA samples from 95 individuals. Each T/S value is the average of triplicate measurements; each mean TRF length is the average of duplicate measurements. Correlation coefficient was equal R2 = 0.844. https://www.ncbi.nlm.nih.gov/pmc/articles/PMC2647324/ 

  1. In the conclusion, the authors stated that the results showed that RTL can be used as a molecular marker as the individual prognosis criterion. The statement over-stated the results. The results only comes from one time point, which cannot rule out the individual variation. The statement needs to be polished. 

Answer: 

We have revised the conclusions:

In this study, we found that the shortage of relative telomere length in patients was significantly correlated with the progression of AMD. The most reduced RTL was observed in patients at an advanced stage of pathology. It was determined that the  significant risk factor for disease progression is RTL < 0.8. The results of our study showed that RTL may be considered as a potential biomarker and a promising predictor of disease progression in patients with early AMD. However, a longitudinal study on a larger sample size with sequential measurements of RTL at different time points is required to evaluate the effect of the rate of telomere length depletion on the rate of disease progression.

Here are some comments directly to the specific point. 

  1. Line 46, the telomere sequence for human should be “TTAGGG”. 

Answer: We corrected that mistake.

  1. Line 48-52 Beside replication error the author mentioned, telomere length also shortens in normal health cells due to the mechanism of DNA polymerase which will not replicate the very end of chromatin. The mechanism is better mention in the paragraph. 

Answer: We corrected that mistake.

  1. Line 126 and line 127 there are two sentences with the same meaning. 

Answer: We corrected lines 126 and 127.

  1. Line 150, the authors stated “… in patients with early AMD can more accurately predict the progression…” the sentence is vague. What marker you are comparing to the telomere length?

Answer: We have revised the sentence: Thus, RTL may be considered as a potential biomarker and a promising predictor of disease progression in patients with early AMD.

Revision of the English language has been carried out

Reviewer 2 Report

In this manuscript, the authors studied the relationship between relative telomere length (RTL) with the risk of development and disease progression. They found out that RTL was significantly lower in AMD patients compare to healthy controls. Women also have a higher change to develop AMD than man does.  In addition, the decrease in RTL is correlated with the progression of AMD, which is a significant finding that can be used as biomarker for early diagnosis of AMD in the future.

1.    The quality of Figure 1 is too low.

2.    They should correct the writing on numbers. For example change 0,99 to 0.99.

Minor changes are required. 

Author Response

We are grateful for your feedback on our article. Below are comments on the corrections made.

  1. According to your comments, we have improved the image figure 1.
  2. In the Russian Federation, it is customary to use a comma to separate the integer and fractional (decimal) parts of a number. Taking into account your remark, we corrected the spelling of numbers and replaced the comma with a dot.
  3. Revision of the English language has been carried out

Round 2

Reviewer 1 Report

I appreciated the improvement from the authors and am satisfied with the current version with only one comment. 

In the Figure 1, the author remove model 4, so the original model 5 became model 4. However, in the according text, there still have model 4 and 5 (assume to be original models?). Please confirm the model numbers are correct and consistent in the text and the figure.

The quality of the English language is significantly improved compared to the original version. 

Author Response

Good afternoon! Thank you for noting the presence of model 5 in the text of the article. In this text fragment, models 4 and 5 refer to models from the supplements, however, in order not to confuse readers, we have changed the text:

“Initially, we calculated models of logistic regression without including telomere length. None of the models without including telomere length were statistically significant. However, models which included such individual covariates as sex, physical inactivity, and cardiovascular disease, showed statistical significance, with the complete regression model remaining statistically insignificant. The highest sensitivity (87%) and specificity (58%) were obtained for the Supplementary Table S1 (Supplementary Figure S1).”